# Vitamin D and Mitosis Evaluation in Endometriosis: A Step toward Discovering the Connection?

**DOI:** 10.3390/biomedicines11082102

**Published:** 2023-07-26

**Authors:** Daniela Roxana Matasariu, Cristina Elena Mandici, Alexandra Ursache, Alexandra Irma Gabriela Bausic, Iuliana Elena Bujor, Alexandra Elena Cristofor, Lucian Vasile Boiculese, Mihaela Grigore, Elvira Bratila, Ludmila Lozneanu

**Affiliations:** 1Department of Obstetrics and Gynecology, University of Medicine and Pharmacy “Gr. T. Popa”, 700115 Iasi, Romania; daniela.matasariu@umfiasi.ro (D.R.M.); cristina-elena_i_tanasa@d.umfiasi.ro (C.E.M.); iuliana-elena.bujor@d.umfiasi.ro (I.E.B.); alexandra-elena_s_mihaila@d.umfiasi.ro (A.E.C.); mihaela.grigore@umfiasi.ro (M.G.); 2Department of Obstetrics and Gynecology, “Cuza Vodă” Hospital, 700038 Iasi, Romania; 3Department of Obstetrics and Gynecology, University of Medicine and Pharmacy “Carol Davila”, 020021 Bucharest, Romania; elvira.bratila@umfcd.ro; 4Department of Obstetrics and Gynecology, “Prof. Dr. Panait Sîrbu” Obstetrics and Gynecology Hospital, 060251 Bucharest, Romania; 5Biostatistics, Department of Preventive Medicine and Interdisciplinarity, University of Medicine and Pharmacy “Gr. T. Popa”, 700115 Iasi, Romania; lboiculese@gmail.com; 6Department of Morpho-Functional Sciences I—Histology, University of Medicine and Pharmacy “Gr. T. Popa”, 700115 Iasi, Romania; ludmila.lozneanu@umfiasi.ro

**Keywords:** 25(OH) vitamin D, vitamin D receptor, phosphohistone H3, progestin, dienogest, endometriosis, immunohistochemistry, diagnosis, risk of endometriosis

## Abstract

(1) Background: The effects of serum vitamin D levels, the vitamin D receptor (VDR), and phosphohistone H3 (PHH3) in endometriosis were investigated in two cohorts of women with this pathology: those receiving hormonal treatment and those without treatment. (2) Methods: In 60 cases of women with endometriosis (26 with progestin treatment and 34 without), paraffin-embedded endometriosis tissue samples retrieved after surgery were immunohistochemically (IHC) analyzed to determine the expression statuses of VDR and PHH3. In addition, serum levels of 25(OH) vitamin D were assessed for each patient. (3) Results: The serum 25(OH) vitamin D evaluations revealed higher levels of 25(OH) vitamin D in women with treatment compared with those without. The positive IHC indexes of VDR and PHH3 in these two groups were compared. Vitamin D receptor levels were positively correlated with PHH3 levels, both being increased in patients without treatment. (4) Conclusions: Serum 25(OH) vitamin D levels and IHC analysis of VDR and PHH3 can be used as additional tools for risk stratification and prognostic assessment in patients with endometriosis.

## 1. Introduction

Due to its severe major impact on women’s health and its incompletely known physiopathology, endometriosis continues to be an important topic of study in the literature. The estrogen-dependent chronic inflammatory disease affects 10 to 15% of women of reproductive age, reaching a higher incidence in cases with infertility or chronic pelvic pain [1,2]. The clinical manifestations of this condition can range from asymptomatic to its primary clinical manifestation consisting of infertility and severe pelvic pain [3,4]. Endometriosis continues to pose a challenge for clinicians, whether they have access to advanced diagnostic tools or not, with a 4- to 11-year delay between the onset of symptoms and a proper diagnosis due to the condition’s nonspecific and wide range of clinical manifestations, as well as the absence of specific tests or biomarkers. The American Society for Reproductive Medicine (ASRM)’s four-stage classification system (stage I—minimum; stage II—mild; stage III—moderate; and stage IV—severe) is the most extensively used classification system for endometriosis [5]. Understanding its underlying causes and potential treatments are the main goals of research on this topic in the literature [1,6]. 

Few studies in the literature have examined vitamin D’s potential therapeutic benefits for endometriosis due to its immunomodulatory, anti-inflammatory, antiproliferative, and anti-invasive properties [6,7,8,9]. Vitamin D levels are categorized as follows: deficiency (<20 ng/mL), insufficiency (20–30 ng/mL), and sufficient level (>30 ng/mL) [7]. The vitamin D nuclear receptor (VDR), which is involved in the transcription process of more than 900 genes, mediates the biological effects of vitamin D. This receptor is engaged in several immunological processes along with the active form of vitamin D, 1,25(OH)_2_ vitamin D. The receptor moves from the cytoplasm into the nucleus after connecting with the active form of vitamin D to initiate gene transcription [10,11].

The PHH3 protein is an essential part of the nucleosome, the fundamental building block of chromatin, and plays a role in the structure and function of DNA [12,13,14]. According to recent investigations, PHH3 alterations may play a role in the onset and progression of endometriosis. For instance, one study discovered that the endometrial tissue of women with endometriosis contained unusually high levels of the PHH3 acetyl group, which may impact the development and survival of abnormal endometrial cells outside the uterus. Additionally, PHH3 acetylation levels in endometriotic lesions were significantly higher than those in healthy endometrial tissue, and this increase has a genetic basis because it is linked to the activation of genes essential for cell survival and proliferation [13,15]. In another study, PHH3 was discovered to be hypermethylated in ectopic endometriotic lesions. This may be a factor in the development of well-known endometriosis characteristics, such as altered gene expression and cellular function [16,17,18,19]. Gujral et al.’s study, for instance, discovered that deacetylating histone through HDAC3 enhanced the progression of endometriosis by controlling the expression of genes linked to inflammation and angiogenesis [18]. Another study that addressed this issue found that JMJD1A, a histone demethylase, was overexpressed in endometriotic tissue and contributed to the onset of endometriosis by controlling the transcription of genes involved in the cell cycle [20]. Moreover, such a good detector of intense cellular proliferation might aid the detection of active endometriotic lesions and, in the future, the elaboration of a mitotic index [14].

Overall, these studies have thoroughly examined the consequences of this protein alteration that may play a part in the pathogenesis of endometriosis, even though the precise role of PHH3 in this disease is still not fully understood [15,16,17,18,19,20]. Additional research is required to better comprehend all the mechanisms and pinpoint further treatment strategies.

The main motivation behind our research was to identify VDR and PHH3 expression in endometriotic lesions to shed light on the physiopathogenesis of this complex and enigmatic disease, as well as to assess serum vitamin D levels to validate its autocrine and/or paracrine actions. Furthermore, we believe these markers represent a promising new direction in the research on the pathogenesis of endometriosis.

## 2. Materials and Methods

### 2.1. Patients and Serum/Tissue Samples 

All patients gave their written informed consent before enrolling in this study, which the Hospital Ethics Committee approved (no. 11062/2020 and no. 6/2021). Formalin-fixed, paraffin-embedded tissue samples were collected from 60 female endometriosis patients who underwent surgery at the Obstetrics and Gynecology Hospital “Cuza-Voda” in Iasi and the Obstetrics and Gynecology Hospital “Panait Sirbu” in Bucharest. The women who were included were all of Caucasian ethnicity. The women proposed for surgical interventions were detected during ultrasound exploration with endometriotic cysts (≥5 cm) and experienced pelvic pain, severe dysmenorrhea and dyspareunia, resistant to medical treatment.

A total of 24 of the 60 patients, ranging from 18 to 45 in age, consented to receive progestin therapy for three months before surgery. The other 36 individuals chose surgical intervention without any prior therapy over hormonal progestin treatment followed by surgical intervention. According to ASRM staging, all the enrolled women had endometriosis at stage III or IV of the disease. Tissue samples were collected between January 2021 and January 2022. We obtained serum samples from a comparable number of patients in each season—6/6/7/7 from those receiving progestin medication and 8/8/9/9 from those without treatment—to eliminate bias resulting from seasonal fluctuations in serum vitamin D levels. The women in our group who had endometriosis but were not receiving treatment were chosen so their ages and BMIs matched those of the patients receiving dienogest. All the samples (serum and tissue) were collected during the menstrual cycle’s proliferative stage. 


*Inclusion criteria:*


We collected tissue and serum samples from women with endometriosis that had been laparoscopically detected and later histologically confirmed. The tissue samples used in the analysis were derived from endometriotic ovarian cysts. We collected specimens from both women who had undergone surgery and those who underwent hormone therapy (with 2 mg of dienogest daily) for three months and from women without any treatment.


*Exclusion criteria:*


We excluded patients with a body mass index (BMI) of >30, malignancy or any other tumoral lesions, diabetes, depression, genetic syndromes, and any infectious or autoimmune disease, as well as women who smoked, pregnant women, those taking any other hormonal therapy besides dienogest or any other treatment that might interfere with bone and mineral metabolisms, and those who took vitamin D supplements for three months before surgery because these factors could have impacted our research.

Histomorphological and immunohistochemical (IHC) testing confirmed the pathological diagnosis in each case. Routinely prepared hematoxylin and eosin (H&E) sections were examined, and the IHC was independently evaluated by two pathologists to confirm the diagnosis.

### 2.2. Serum Samples

Serum levels of 25(OH) vitamin D were assessed using an electrochemiluminescent assay (ECLIA), the Roche Elecsys 2010 Rack Immunology Analyzer, provided by Roche Diagnostics. All serum samples were stored at −80 °C until the analysis was performed.

### 2.3. Immunohistochemistry

IHC staining was performed on tissues that had been formalin-fixed and paraffin-embedded.

Four-micrometer-thick serial sections on coated sections were prepared in citrate buffer (pH 6) following deparaffinization in xylene and rehydration in an ethanol series. H_2_O_2_ of 0.3% was used to inhibit endogenous peroxidase activity for 20 min at room temperature. Vitamin D antireceptor was diluted at 1:3000 (Abcam, Cambridge, UK) and incubated overnight at 4 °C, and the monoclonal anti-PHH3 antibodies’ (BioSB, Santa Barbara, CA, USA) dilution was 1:250. The sections were washed, exposed to the secondary antibody for 45 min at 37 degrees, and then thoroughly cleaned with *phosphate-buffered saline (*PBS*)* (Table 1). Hematoxylin was used as a counterstain in the standard avidin–biotin–peroxidase technique, in which a liquid DAB (diaminobenzidine) substrate and chromogen system was used for viewing. Primary antibodies were left out as a negative control. As positive references, human jejunum was utilized for VDR and human lymph node tissue section for PHH3.

Sixty samples with endometriosis in the tissues’ stromal and epithelial components were examined for VDR and PHH3. Patients who received treatment were compared with those without treatment. 

Positive cells (indicated by a brown or yellowish-brown color in the nucleus) in the epithelial and stromal compartment areas under a light microscope were characterized as VDR- and PHH3-positive. Any stained nuclear pattern was considered positive regardless of the staining intensity or the number of positive cells in the stromal or epithelial compartments.

### 2.4. Statistical Analysis

Medical data were imported and verified with Microsoft Excel and then analyzed using SPSS 24 (IBM Corp. Released 2016. IBM SPSS Statistics for Windows, Version 24.0., Armonk, NY, USA: IBM Corp.).

The information was in the form of measurable numerical values on a ratio or interval scale as categorical variables. We calculated the values of the following statistical measures within the descriptive statistics: sample size (N), mean, standard deviation, standard error, and the mean, min., max., absolute, and relative frequencies with a 95% confidence interval.

Statistical hypothesis tests were performed using *t*- or Student’s *t*-tests (to compare the averages of numerical variables), the Levene method (to check the equality of variances), and chi-square or Fisher’s exact tests (for categorical type).

The standard cut-off of 5% or 0.05 significance was used to decide the conclusion of the hypothesis.

## 3. Results

Our included patients’ ages ranged from 18 to 45 years, with a mean age of 31.92 ± 4.706 (95% CI 30.70/33.13; std error 0.608).

Before surgery, 24 (40%) patients underwent progestin treatment with dienogest, while 36 (60%) patients received no treatment at all. 

The 25(OH) vitamin D serum levels are presented in Table 2. We detected no statistically significant differences between the two groups of women with endometriosis, those with treatment and those without treatment, which concerned serum vitamin D levels, but women under treatment with dienogest had higher 25(OH) vitamin D serum levels (*p*–0.287).

We tried to establish the patterns of expression and distribution of the VDR and PHH3 to better understand their roles in ovarian endometriotic cells.

In this study, 20 women (83.3%) of the 24 included in the treatment group were VDR-positive and only 4 (16.7%) were negative. In the group of women with endometriosis but without treatment, 32 (88.9%) were positive, and only 4 (11.1%) were negative (Table 3).

In the analysis of PHH3 expression in the treatment group, 8 women (33.3%) were found positive and 16 (66.7%) negative, and in the group without treatment, 20 (55.6%) were positive and 16 (44.4%) were negative (Table 3).

The expression levels of VDR in the group without treatment at 32 (88.9%) were higher than those in the group with treatment at 20 (83.33%). The negative expression of VDR was similar in both groups (Table 3). 

VDR showed diffuse positive nuclear expression in both epithelial and stromal (Figure 1A–D) compartments; however, the VDR expression levels in the epithelial compartment in the treated group tended to be lower at 20 (35.71%) than those in the untreated group at 36 (64.28%) (Table 3 and Table 4).

In contrast with the negative expression of PHH3, which was similar in both groups, the expression levels of PHH3 in the group without treatment at 20 (55.6%) were higher than those in the group with treatment at 8 (33.3%) (Table 3). 

PHH3 was expressed in both endometriotic areas, epithelial and stromal compartments (Figure 1E–H), irrespective of therapy status. We noticed that patients receiving medication had lower levels of PHH3 expression in the epithelium, with 12 (30%) cases, than patients not receiving treatment, with 28 (70%) cases (Table 4).

Comparative images of the immunohistochemical staining in the two groups (with and without treatment).

## 4. Discussion

According to the findings of our study, endometriosis patients who received dienogest treatment for three months before surgery had higher serum levels of 25(OH) vitamin D. The fact that the medication had such a quantitative effect after three months of therapy, despite the difference not being statistically significant, suggests a strong relationship between this disease and circulating vitamin D. 

Contradictory results have been found in human investigations even though in vitro and animal studies have supported the immunomodulatory effects of vitamin D in endometriosis, with the regression of implants. The extremely varied nature of endometriosis may be a factor in the inconsistent nature of study findings. Additionally, it is challenging to accurately measure 1,25(OH)_2_ vitamin D, the active form of vitamin D, due to its short half-life [7,8,21,22]. Agic et al. have proposed that vitamin D, in addition to its well-known endocrine function, may have more autocrine and/or paracrine local effects that contribute to endometriosis. The researchers detected higher vitamin D levels in ovarian endometriotic cysts compared with normal ovarian tissue. We measured VDR expression in tissue samples (both in the epithelium and stroma) from women with endometriosis to take a further step in the direction suggested by Agic et al., 2007 [23].

Most studies indicate either statistically significant or non-statistically significant lower 25(OH) vitamin D serum levels in women with endometriosis [7,9,22,23,24]. Our motivation for proceeding with our research and examining whether progestin treatment affects serum vitamin D levels and endometriotic ectopic tissue samples was the hypothesis that vitamin D supplementation associated with specific endometriosis treatment might lead to an improvement in this disease’s evolution in patients. This is because the majority of studies in the literature have focused on vitamin D’s effect of reducing the primary symptom of endometriosis, dysmenorrhea. Our approach was to assess both serum vitamin D, which promotes secretion and regulates normal cellular growth at a local level, and its receptor’s expression in ectopic endometriotic ovarian tissue, focusing on the role of vitamin D in defining endometriosis [25,26].

There is scarce research that has analyzed the tissue levels of VDR in endometriosis-affected women. Due to numerous discrepancies found in human and animal studies about serum vitamin D levels and their effects on endometriosis patients, we decided to conduct this analysis at the cellular level. In another investigation, Miyashita et al. found that rather than having a proapoptotic effect, vitamin D treatment not only reduced inflammation but also significantly decreased the viability of human endometriotic stromal cells [24]. According to our research, progestin therapy and dienogest therapy reduce VDR expression in the stromal and epithelial compartments of ectopic endometriotic samples, respectively, with the epithelial compartment displaying a more pronounced decrease. Therefore, hormonal progestin therapy appears to have a positive local impact. Another aspect of vitamin D’s role in endometriosis is indicated by progestin treatment’s direct impact on VDR expression. This aspect, along with low serum vitamin D levels, high VDR endometriotic ectopic cell expression, and increased vitamin D serum levels in progestin-treated endometriosis patients, suggests that vitamin D supplementation is required in patients with this condition to lessen the severity and spread of the illness [7,9,24,25,26,27].

In their 2021 investigation, Yarmolinskaya et al. considered the polymorphism of the vitamin D nuclear receptor while evaluating the expression of VDR in the eutopic and ectopic endometrium in 32 women with endometriosis and 20 healthy controls [22]. To confirm the favorable effects of hormonal treatment on this enigmatic disease, we intended to examine the receptor’s presence in endometriosis-affected women who were or were not receiving progestin treatment. The researchers discovered no cyclic fluctuations in the menstrual cycle phases compared with controls and lower VDR expression in the ectopic endometrium than in the eutopic one in women with endometriosis. The researchers further investigated this area by examining VDR expression in endometriotic cyst specimens from women who had received endometriosis diagnoses and contrasting women with treatment with those who had not received treatment. High amounts of VDR were found in the samples collected from women who had not received treatment [22]. The increased expression of VDR in the endometrium of endometriosis-suffering women without therapy shows that the molecular pathway is actively involved in the development and pathophysiology of endometriosis. Nevertheless, there is a strong correlation between hormonal action and the low expression of VDR in the group receiving treatment.

We are unaware of other studies on VDR expression in women’s endometriotic ectopic tissues beyond the aforementioned Russian study [22,23,28,29].

Vitamin D functions as a cellular signal transducer to suppress proliferation in the group receiving treatment due to its lower receptor expression in epithelial and stromal components [30,31]. Based on their findings, the authors of an extensive review of the available literature on vitamin D from 2018 concluded that vitamin D is a steroid hormone with progesterone-like activity. These findings, along with ours, suggest that VDR may be linked to some endometrioses’ aggressiveness in individuals who are not receiving treatment [31]. Additionally, by highlighting the antifibrotic properties of vitamin D, Monastra et al. suggest a new line of inquiry: investigating whether vitamin D supplementation, in addition to standard progestin therapy, might help endometriosis patients avoid the fibrosis and adhesions that are characteristic of their condition and alleviate pain-related symptoms [32].

The PHH3 molecule is a mitosis marker, and its nuclear-modified expression has been studied in different types of cancers. Studies state that it might serve as an indicator of poor prognosis in such patients. Because of their specific traits, endometriotic cells behave similarly to those in malignancies, displaying reduced apoptosis, intense proliferation, low cell adhesion, and invasivity. We assumed that progestin therapy might be a factor that inhibits the proliferation of endometriotic cells and that, as a result, PHH3 expression in treated patients would be lower than in untreated women [33].

Information about PHH3 expression in endometriosis remains scarce. The PHH3 protein, which is known to be involved in DNA-templated reactions to the nucleosome, is thought to undergo several genetic alterations over the course of this disease’s development and progression [2,18].

It is hypothesized that PHH3, which functions as a nucleosome for DNA synthesis, plays a role in endometriosis patients’ increased proliferation and decreased apoptosis rates in the epithelial and stromal areas. In endometriosis patients, PHH3 knockdown contributes to preventing cellular division and proliferation, which raises the apoptosis rate in the epithelial and stromal endometriosis areas [16,19]. According to these findings, the current investigation revealed that a significant percentage of the group not receiving therapy expressed PHH3.

We discovered that patients receiving treatment had lower levels of PHH3 and VDR, indicating a protective effect of the hormone therapy. PHH3’s role in the risk stratification and prognosis of endometriosis is related to direct protein expression. Additionally, there appears to be a direct relationship between PHH3 and VDR, whereby as PHH3 expression rises, indicating intensified mitotic activity, VDR expression also rises, indicating an active endometriotic lesion. The use of PHH3 IHC analysis facilitates the detection of endometriosis for clinicians.

### Study Limitations

Our research has some limitations. The first is the small number of patients whose tissue samples were examined and included in our analysis. Extensive additional studies are needed in this area to understand the variation in previous findings on vitamin D levels in endometriosis patients. The second limitation is that only endometriotic cysts were used for sample collection in all cases. This disease phenotype cannot be regarded as typical of an illness as complex as endometriosis. To explore this further, we will try to correlate these results with serum vitamin D and PHH3 levels in these patients.

## 5. Conclusions

To the best of our knowledge, this is the first study to evaluate tissue VDR expression in women with endometriosis with and without treatment, demonstrating an increase in circulating vitamin D levels in women with endometriosis under progestin treatment. VDR overexpression was linked to patients without treatment, displaying low 25(OH) vitamin D serum levels. PHH3 overexpression was associated with the group that received no therapy, and additional analysis proved it to be statistically significant, although non-specific. 

Additionally, to our knowledge, the relationship between PHH3 and VDR tissue expressions in endometriosis-affected women was analyzed for the first time in this study.

By showing decreased serum vitamin D levels in endometriosis-affected women, along with raised tissue vitamin D receptor levels in endometriotic specimens, we were able to demonstrate a connection between the two. If we can continue to show that vitamin D supplementation has noticeable beneficial benefits on endometriotic lesions, we will have a new course of therapy to advocate for this debilitating condition. On the other hand, vitamin D may be suggested as a therapeutic alternative for women who wish to conceive. There are not many additional hormonal treatment options accessible for this group of women. If additional research validates the changes in PHH3 levels, we can also use it as a non-invasive diagnostic marker or a marker to monitor the efficacy of treatment that might prove to be more accurate than CA 125.

## Figures and Tables

**Figure 1 biomedicines-11-02102-f001:**
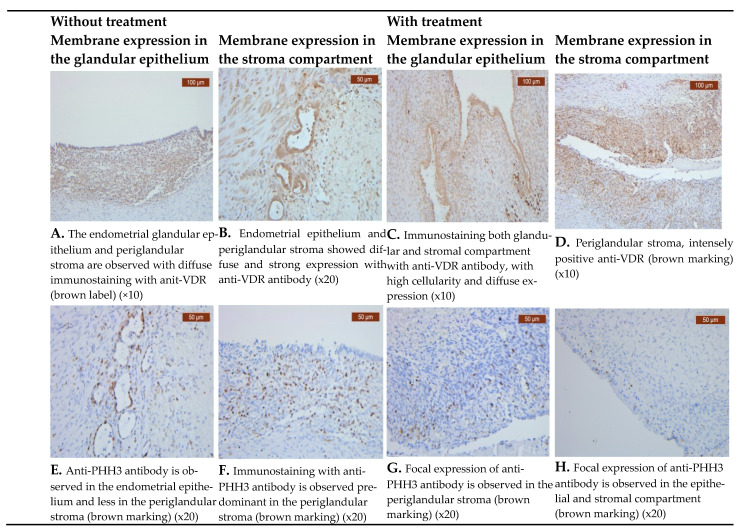
Immunohistochemical expression of VDR and PHH3 in two groups (with and without treatment).

**Table 1 biomedicines-11-02102-t001:** Immunohistochemical panel of antibodies used in our study.

Antibody	Clone, Manufacturer	Dilution	Expression
Anti-PHH3	EP233, rabbit monoclonal IgG isotype, BioSB	1:250	Nuclear
Anti-VDR	Rabbit polyclonal IgG isotype, Abcam	1:3000	Nuclear

PHH3: phosphohistone H3; VDR: vitamin D receptor; IgG: Immunoglobulin G.

**Table 2 biomedicines-11-02102-t002:** Serum vitamin D levels: descriptive statistical characteristics.

Group	N	Mean	Std. Deviation	Std. Error	95% ConfidenceInterval for Mean		Min.	Max.	*p*
					Lower Bound	Upper Bound			
Without treatment	36	17.053	2.446	0.407	16.225/17.881		11.9	20	0.287
With treatment	24	16.279	3.111	0.635	14.965/17.593		10.3	20	

**Table 3 biomedicines-11-02102-t003:** Differential expressions of VDR and PHH3 in endometriotic samples.

Group	N	VDR		PHH3	
		Positive	Negative	Positive	Negative
With treatment	24	20 (83.3%)	4 (16.7%)	8 (33.3%)	16 (66.7%)
Without treatment	36	32 (88.9%)	4 (11.1%)	20 (55.6%)	16 (44.4%)
*p*-value		0.702 F		0.091	

**Table 4 biomedicines-11-02102-t004:** Differential expressions of VDR and PHH3 in endometriotic samples.

		VDR				PHH3			
Group	N	Epithelial		Stromal		Epithelial		Stromal	
		Positive	Negative	Positive	Negative	Positive	Negative	Positive	Negative
With treatment	24	20 (35.71%)	4 (100%)	16 (40%)	8 (40%)	12 (30%)	12 (60%)	16 (50%)	8 (28.57%)
Without treatment	36	36 (64.28%)	0 (0%)	24 (60%)	12 (60%)	28 (70%)	8 (40%)	16 (50%)	20 (71.42%)
*p*-value		56	4	40	20	40	20	32	28
		0.02		1		0.025		0.09	

## Data Availability

The data used to support the findings of this study are available upon request from the corresponding author.

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
