# Peer review of "Vitamin D and Mitosis Evaluation in Endometriosis: A Step toward Discovering the Connection?"

_biomedicines, 2023, doi:10.3390/biomedicines11082102_

Round 1

Reviewer 1 Report

The manuscript "Vitamin D and mitosis evaluation in endometriosis – a step in discovering the connection?" is an interesting manuscript on the role of vitamin D in endometriosis. The work is complete and well structured, giving important data to scientific literature. The design of the project is appropriate and the results are significant. The statistical analysis is well conducted and the language is acceptable. The main concern is the clinical utility of the data of this work:  what are the actual clinical implications of this study? it is important to report the results obtained by the authors in the context of clinical practice and to adequately highlight what contribution this study adds to the literature already existing on the topic and to future study perspectives. It represents a valid work and it gives the opportunity to focus attention on possible future applications of Vitamin D in Endometriosis

Author Response

Best regards,

Alexandra Ursache, MD, PhD

Reviewer 2 Report

Thanks to the authors for their work.

The title looks good.

The abstract needs revision in language and organization. All statistics need to be demonstrated.

Keywords: looks good, you may add “diagnosis” or “risk of endometriosis”

The introduction: the first 66 lines need to condense into one paragraph and more focus on the problem addressed.

Line 135: why the surgery was done?

Line 144: are these endo patients? If yes why no treatment? What are the participants’ clinical features?

Please provide the clinical symptoms of patients and correlate them with your results

The discussion section needs a lot of organization and rewriting

The Russian work needs to be cited in the text.

The first paragraph of the discussion should be about you’re finding.

Please check reference 23. It is not Russian??

THe language needs revisions

Author Response

(The authors gave the same response as above.)

Round 2

Reviewer 2 Report

many thx for your reply and corrections. The manuscripts look much better with the exception of minor typo mistakes, that need revision.

 The manuscripts look much better with the exception of minor typo mistakes, that need revision. Please revise the language.